# Biocatalytic trifluoromethylation of unprotected phenols

Robert C. Simon[1], Eduardo Busto[1], Nina Richter[2], Verena Resch[1], Kendall N. Houk[3] & Wolfgang Kroutil[1]

Organofluorine compounds have become important building blocks for a broad range of advanced materials, polymers, agrochemicals, and increasingly for pharmaceuticals. Despite tremendous progress within the area of fluorination chemistry, methods for the direct introduction of fluoroalkyl-groups into organic molecules without prefunctionalization are still highly desired. Here we present a concept for the introduction of the trifluoromethyl group into unprotected phenols by employing a biocatalyst (laccase), $t$BuOOH, and either the Langlois' reagent or Baran's zinc sulfinate. The method relies on the recombination of two radical species, namely, the phenol radical cation generated directly by the laccase and the $CF_3$-radical. Various functional groups such as ketone, ester, aldehyde, ether and nitrile are tolerated. This laccase-catalysed trifluoromethylation proceeds under mild conditions and allows accessing trifluoromethyl-substituted phenols that were not available by classical methods.

[1] Department of Chemistry, Organic and Bioorganic Chemistry, University of Graz, NAWI Graz, BioTechMed Graz, Heinrichstrasse 28, 8010-Graz, Austria. [2] ACIB GmbH, c/o Heinrichstrasse 28, 8010-Graz, Austria. [3] Department of Chemistry and Biochemistry, University of California, Los Angeles, California 90095, USA. Correspondence and requests for materials should be addressed to R.C.S. (email: robert.simon@gmx.net) or to K.N.H. (email: houk@chem.ucla.edu) or to W.K. (email: wolfgang.kroutil@uni-graz.at).

The introduction of fluoroalkyl-groups (for example, CF₃, CHF₂, CH₂F, etc.) into organic compounds has become a major subject in various fields of chemical research, in particular medicinal chemistry and drug discovery[1,2]. This is due to the metabolic stability, increased permeability or enhanced binding properties of the organo-fluorine compounds in comparison to their non-fluorinated counterparts[3]. Among all fluorine-containing moieties, the trifluoromethyl group is privileged[4], and trifluoromethylated arenes are of interest for agrochemicals, pharmaceuticals and advanced materials[5]. Several excellent methods to provide structurally diverse CF₃-building blocks have been elaborated[6–8]: common strategies to introduce the CF₃-group into aromatic compounds involve metal-mediated/catalysed functional group interconversions[9,10] where halogens[11–13], boronic acids[14–16], boronates[17,18] and even amines[19,20] are replaced by nucleophilic, electrophilic or radical CF₃-sources (Fig. 1a). Other methods rely on directing groups[21–23], as well as on visible light and photo-catalysis (Fig. 1b,c)[24–26].

While trifluoromethylation of (substituted) mono- and biaryl-systems has been broadly investigated[27–30], only few reports deal with the transformation of unprotected phenols (Fig. 1d)[31–33], and these give non-regioselective transformations and/or unsatisfying conversions. Hence, a general method for attaching the CF₃-moiety to phenols in a practical manner remains elusive. We report an efficient and selective method for trifluoromethylation of unprotected phenols by biocatalytic introduction of a trifluoromethyl group derived from common precursors.

## Results

**Reaction concept.** The approach for the trifluoromethylation of phenols via C–C bond formation presented in this paper is based on the recombination of two radicals, namely a CF₃-radical and a phenol-derived radical, wherein the two radicals are formed via two different pathways (Fig. 2). The phenol-derived radical is formed by a laccase (E.C. 1.10.3.2), which catalyses in general the one-electron oxidation of phenols and anilines using molecular oxygen as the oxidant[34–38]. Simultaneously, the electrophilic CF₃-radical is generated *in situ* from either Langlois' reagent (NaSO₂CF₃)[31,39,40] or Baran's zinc sulfate

(Zn(SO₂CF₃)₂ = TFMS)[41]. The CF₃-radical may also be formed under electrochemical conditions from TFMS[42].

The concept was initially tested by employing the laccase from *Agaricus bisporus* and using the electron-rich phenol **1a** as substrate and TFMS as trifluoromethylation agent with *tert*-butyl hydroperoxide (*t*BuOOH) as the oxidant. This ortho- and para-substituted substrate was chosen to minimize literature-known fast di- or oligomerization in ortho and para positions initiated by laccases[34–37,43]. The chemical-enzymatic system led to successful product formation, whereby O-alkylation was not detected, but C–C bond formation at the free C–H of the arene was observed, leading to product **2a** with 58% of the transformed substrate (Table 1, entry 1).

The phenol did not react in the absence of laccase and in the presence of TFMS and *t*BuOOH (Entry 2). In the presence of the laccase but in the absence of TFMS/*t*BuOOH, substrate **1a** was just di-/polymerized (Entry 3). Also, performing the reaction with laccase and TFMS in the absence of *t*BuOOH did not lead to the desired product formation of **2a**, indicating that *t*BuOOH is required to form the CF₃-radical (Entry 4). Thus, product **2a** was only formed in the presence of laccase and TFMS/*t*BuOOH. *t*BuOOH cannot be substituted by hydrogen peroxide, since in this case neither the formation of trifluoromethylated **2a** nor any other transformation of **1a** was observed (entry 5).

Encouraged by this initial result, the radical trifluoromethylation of **1a** was optimized, testing varied concentrations of TFMS, *t*BuOOH, co-solvents as well as temperature (see Supplementary Information). The best conditions for the formation of **2a** were found at 50 mM phenol **1a** with 2.0 eq. TFMS and 8.0 eq. *t*BuOOH at 30 °C in the presence of 25 vol% dimethylsulfoxide (DMSO).

**Functional group tolerance.** To tap the scope and functional group tolerance of this method, various substituted phenols **1a-h** were transformed under optimized conditions, whereby the ortho- and para-position with respect to the phenolic hydroxyl moiety were blocked for substrate **1a-c** (Table 2). In these cases, products **2a-c** with the CF₃-moiety *meta* to the OH were isolated with exquisite regio-control (entry 1–3), and for **2b** and **2c** also verified by X-ray crystallography (Fig. 3). Comparable results were obtained independent of the trifluoromethylation agent employed: thus, the Langlois' reagent as well as the Baran's zinc sulfate led to comparable isolated yields up to 62%. Moreover, the reaction system tolerated ketone-, ester-aldehyde, as well as nitrile-functionalities, emphasizing the mildness of the reaction. Interestingly, nitrogen-containing substrates like indol or 4-aminoacetophenone were not converted at all under the reaction conditions investigated, while other substrates like

## Figure 1

**Figure 1 | Methods to attach the CF₃-group to arenes.** (**a**) Functional group interconversion; (**b**) Trifluoromethylation controlled by a directing group (DG); (**c**) Innate trifluoromethylation of substituted arenes, accept phenols; (**d**) Innate trifluoromethylation of phenols.

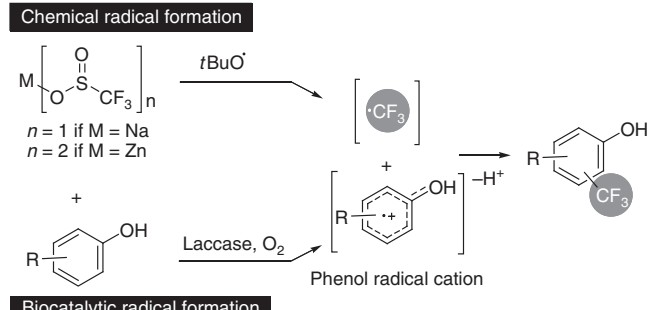

**Figure 2 | Biocatalytic trifluoromethylation of unprotected phenols.** Radical trifluoromethylation of unprotected phenols by recombination of radicals generated via two different pathways.

**Table 1 | Laccase-mediated trifluoromethylation of unprotected phenols in combination with TFMS.**

| Entry | Catalysts/reagents | Conv. (%)* | 2a (%)† |
|---|---|---|---|
| 1 | Laccase + **1a** + TFMS + $t$BuOOH | $99.4 \pm 0.2$ | $58.0 \pm 1.7$ |
| 2 | TFMS + $t$BuOOH + **1a** | n.c. | — |
| 3 | Laccase + **1a** | $77.1 \pm 0.3$ | $<0.1$ |
| 4 | Laccase + **1a** + TFMS | $23.6 \pm 0.8$ | $<0.1$ |
| 5 | Laccase + **1a** + TFMS + $H_2O_2$ | n.c. | — |

Laccase, *Agaricus bisporus* laccase; n.c., no conversion;
TFMS = Zn(SO$_2$CF$_3$)$_2$ = (((trifluoromethyl)sulfinyl)oxy)zinc salt.
*Based on recovered starting material.
†Determined by GC.

sesamol, 5,6,7,8-tetrahydro-2-naphthol, 2-naphthol or meta-dimethylamino acetophenone resulted in complex product mixtures.

**Regioselectivity.** The transformation of phenols **1d** and **1e**, bearing only a single substituent ortho to the phenolic hydroxy group, led to a mixture of regio-isomers, albeit with significant preference for the meta-isomers (C2:C3 = 4:1 up to 10:1, entry 4 and 5). The preferred meta-substitution for **2d** was confirmed via a crystal structure (Fig. 3). It is worth noting that substrate **1f**, being devoid of ortho-methoxy substituents, afforded only the isomer bearing the CF$_3$ moiety ortho to the alcohol group with 31% isolated yield (entry 6). In a similar fashion, substrate **1g** possessing in para position a nitrile group instead of the acetyl moiety resulted in the mono-substituted ortho-product **2g** with 57% isolated yield (entry 7). Taking a substrate devoid of a para-substituent but having methoxy substituents in both ortho positions (**1h**), the trifluoromethylation protocol led to a di-trifluoromethylated product, **2h**, having a CF$_3$-group in ortho as well as meta position with respect to the phenolic OH (entry 8).

The observed regioselectivity for **2d**-**f** can be explained by transition state energies of the addition of the CF$_3$-radical to the phenol radical cations. For instance, the corresponding transition state leading to **2d** with substitution at C2 is energetically preferred over substitution at C3 (2.1 kcal mol$^{-1}$, M06-2X/6-311 + G(d,p)) (Fig. 4, Supplementary Fig. 5). The same is true for the analogous transition state leading to **2e** (Supplementary Fig. 6). The energies of the substituted intermediate cations after addition of the CF$_3$-radical to the phenol radical cation also reflect the observed regioselectivity. In the case of substrate **1f**, the energies of the transition states support the expected and observed substitution in ortho-position to the phenolic OH leading to product **2f** (Supplementary Fig. 6). Since the energies of the transition states reflect the observed regioselectivity, the bio-trifluoromethylation is mainly not active site-directed.

**Mechanism.** The computational calculations also showed that the CF$_3$-radical has to react preferentially with the phenol radical cation and not with the corresponding already deprotonated phenoxy radical, since the latter would lead, for example, for **1d**, to substitution in ortho-position to the hydroxy group and not at the mainly observed meta-position. Substitution in ortho-position

**Table 2 | Scope and functional group tolerance of the biocatalytic trifluoromethylation of unprotected phenols employing Baran's zinc sulfinate or Langlois' reagent in combination with a laccase.**

| Entry | Product | Zn(SO$_2$CF$_3$)$_2$ Isolated. yield (%) | NaSO$_2$CF$_3$ Isolated yield (%) |
|---|---|---|---|
| 1 | **2a** | 61.6 | 52.5 |
| 2 | **2b [x-ray]** | 42.2* | 40.5* |
| 3 | **2c [x-ray]** | 57.8 | 57.3 |
| 4 | **2d [x-ray]** | 41.5† (C2:C3:C6 = 10:1:1) | 29.2† (C2:C3 = 4:1) |
| 5 | **2e** | 37.2†† (C2:C3 = 4:1) | 31.3†† (C2:C3 = 4:1) |
| 6 | **2f** | 31.7 | 30.5 |
| 7 | **2g** | 56.7 | n.p. |
| 8 | **2h** | 33.1 | n.p. |

n.p. = not performed.
* ∼4% double trifluoromethylation at C2 and C6 was observed.
†Isolated yield of the pure regio-isomer **2d**-*meta*.
‡Overall isolated yield of both regio-isomers **2e**-*ortho* and **2e**-*meta*.

would be favoured over meta-position by 3 kcal mol$^{-1}$ upon combination with the neutral phenoxy radical. Therefore, in the proposed mechanism the laccase oxidizes the phenol **1d** via a single electron transfer to the phenol radical cation (Fig. 5). The latter reacts with the CF$_3$-radical to give the cationic intermediate, which rearomatizes to the final product. As shown in the initial experiments, trifluoromethylation only occurred in the presence of laccase and TFMS/$t$BuOOH (Table 1, entry 1); the phenol

starting material did not react with TFMS/tBuOOH (entry 2), nor did the phenol radical cation (formed by laccase and $O_2$ present) react with TFMS (entry 4).

In the recently proposed mechanism[39], traces of redox metals are proposed to initiate the reaction for the first transformation of tBuOOH to tBuO• and $OH^-$. tBuO• enables the formation of the $CF_3$• species. In the reported catalytic cycle the activation of tBuOOH was triggered by the heteroaromatic radical intermediates.

Since in the laccase-catalysed trifluoromethylation of phenol, the laccase provides already one reactive radical species for the C–C bond forming reaction, namely the phenol radical cation, stoichiometric amounts of redox metal (for example, Fe, Co, Cu mentioned in previous work) would be required to obtain the amount of tBuO• needed. Since the trifluoromethylation went to high conversion without addition of any metals or other redox reagents, it was deduced that the copper Cu(I) present in the laccase also reacts with tBuOOH to give tBuO• and Cu(II) as already proposed in previous papers using only Cu (ref. 31). This

was also supported by photometric assays in laccase-catalysed oxidative dimerization of 2,6-dimethoxy phenol showing that the presence of tBuOOH led to a faster reaction (Supplementary Methods, Photometric Enzymatic Activity Assay). Thus, the Cu(I) of the laccase can be oxidized by tBuOOH, which leads to tBuO• as previously reported[31]; the latter reacts with the $CF_3SO_2^-$ to set free the $CF_3$-radical, as proven elsewhere[39].

**Comparison to literature methods.** To compare the here-presented laccase/tBuOOH protocol with published methods for the chemical trifluoromethylation of (electron-rich) arenes and hetero-arenes[29,30], phenols **1a** (R = Me) and **1b** (R = H) were treated with Ruppert-Prakash reagent[44] $TMSCF_3$ in the presence of catalytic silver (AgF) and $PhI(OAc)_2$ as oxidant (Fig. 6a).

For both substrates **1a** and **1b** only minor amounts of **2a** and **2b** were found in a complex product mixture; the major product components were the trifluoromethyl-aryl-ethers **3a** and **3b** (13–17% isolated yield). A related $CF_3$-ether formation was reported recently[45].

As a second literature method a metal-free alternative for the trifluoromethylation of arenes and biaryls was investigated[30], whereby the $CF_3$-radical is generated by the oxidation of the Langlois' reagent $NaSO_2CF_3$ with phenyl-iodine bis(trifluoro-acetate) (Fig. 6b). In this case the transformation of **1a** and **1b** led to the corresponding trifluoromethanesulfonates **4a** and **4b** as the main products (18–89% isolated yield), while **2a** and **2b** (R = H) were found only in negligible quantities.

As a third method the Togni reagent[32,46] was employed for substrate **1a** (Fig. 6c); although the substrate was completely converted, product **2a** was only a minor product (8%), while two non-identified main products were detected, which did not contain any $CF_3$-group.

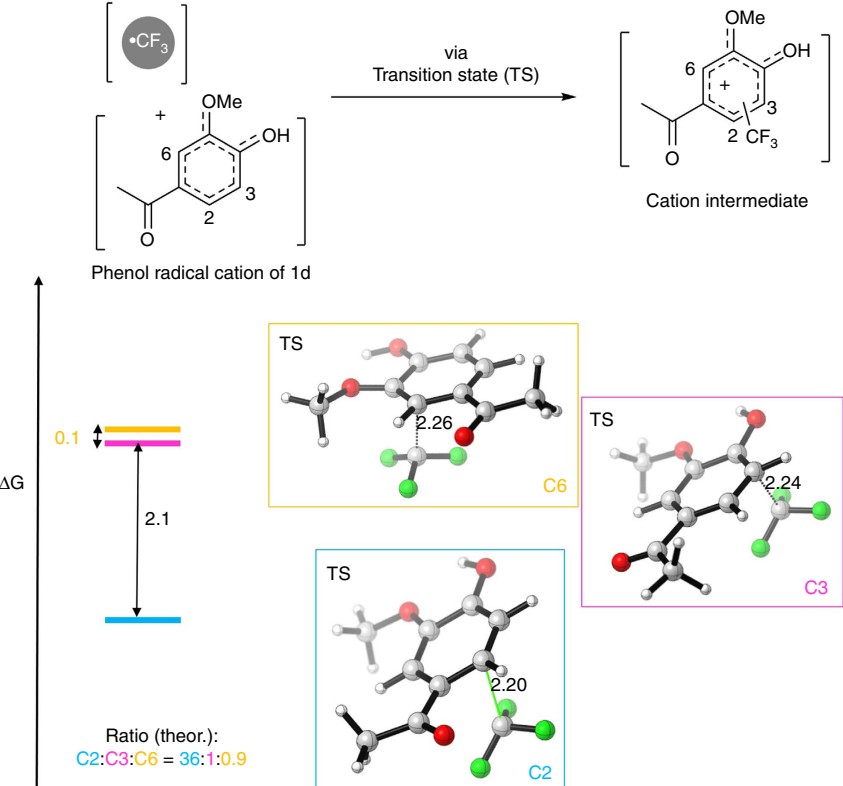

**Figure 3 | Crystal structures of 2b, 2c and 2d-meta.** Stereoscopic ORTEP plot of crystals **2b**, **2c** and **2d**-meta. The probability ellipsoids were drawn on 50% probability. Radii of hydrogen atoms are drawn arbitrarily.

Ratio (theor.):
C2:C3:C6 = 36:1:0.9

**Figure 4 | Energy differences of transition states.** Energy differences and structures of transition states leading preferentially to substitution at C2 giving **2d**.

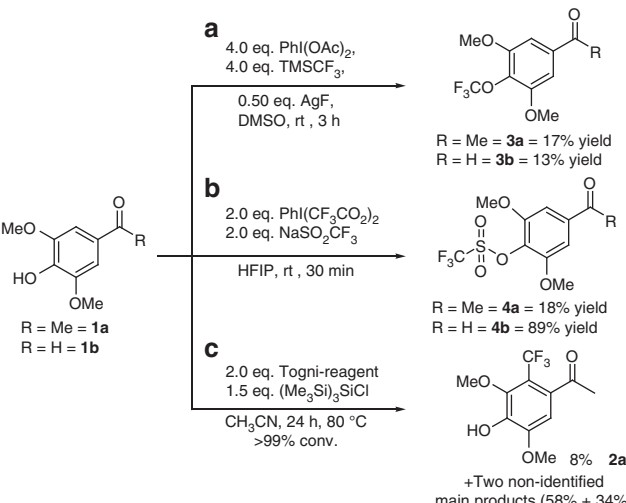

**Figure 5 | Proposed mechanism for laccase-catalysed trifluoromethylation of phenols.** Proposed mechanism for the laccase-mediated trifluoromethylation of unprotected phenols exemplified for substrate **1d**. For reasons of clarity the scheme displays only the productive pathway relevant for the formation of **2b**; other reactions, for example, like $HCF_3$ formation, dimerization of the $CF_3$-radical or the radical cation, the decay of $t$BuOOH and the oxidation of $SO_2$ are omitted, as well as the formation of minor regio-isomers.

## Methods

**Representative trifluoromethylation procedure (1 ml).** The laccase from *A. bisporus* (7.5 U, 5.0 mg ml$^{-1}$ final conc.) was dissolved in a sodium acetate buffer (695 µl, 250 mM, pH 5.5) prior to addition of $Zn(SO_2CF_3)_2$ (2 eq., 33.2 mg dissolved in DMSO). Afterwards ketone **1** (50 mM final concentration, dissolved in DMSO) was added followed by aqueous $t$BuOOH solution (8.0 eq., 55 µl, 70 wt% aqueous solution) to reach a total volume of 1.0 ml (25 vol% DMSO). The reactions were shaken in an orbital shaker at 30 °C ($Zn(SO_2CF_3)_2$) or 40 °C in case of $NaSO_2CF_3$ for 24 h at 900 r.p.m. (horizontal position). Then, each 1 ml reaction was extracted four times with EtOAc (500 µl) and combined organic fractions were dried over $Na_2SO_4$. The solutions were filtered, concentrated under reduced pressure and the residue was purified by various solvent mixtures to afford the trifluoromethylated phenol derivative **2**.

**QM calculations.** Full geometry optimizations, transition structure searches and single-point computations were carried out with the Gaussian 09 package[47]. All geometry optimizations were carried out with the unrestricted version of the hybrid B3LYP functional[48]. For C, O, N and H, the double-zeta basis set 6–31G(d) was employed to obtain the geometries, and the larger 6–311 + G(d,p) basis set was used to calculate single-point energies. Additional single-point energy calculations using functionals able to account for dispersion forces such as M06-2X (ref. 49) in conjunction with the 6–311 + G(d,p) basis set were performed (Supplementary Table 5). Thermal and entropic corrections to energy were calculated from vibrational frequencies. The nature of the stationary points was determined in each case according to the appropriate number of negative eigenvalues of the Hessian matrix from the frequency calculations. Frequencies were not scaled.

**Data availability.** Crystal structures that support the findings of this study have been deposited at the Cambridge Crystallographic Data Centre and allocated the deposition numbers CCDC 1480621 (**2b**), 1480623 (**2c**) and CCDC 1480622 (**2d**). All other data supporting the findings of this study are available within the article and its Supplementary Information file or from the author upon reasonable request.

**Figure 6 | Methods from literature tested for comparison.**
Trifluoromethylation methods for electron-rich arenes for comparison with the here-presented laccase/$t$BuOOH concept. (**a**) Method involving silver as metal; metal-free trifluoromethylations using phenyl-iodine bis(trifluoroacetate) (**b**) or Togni-reagent (**c**).

Thus, the laccase trifluoromethylation reported here is clearly complementary to literature methods tested.

## Discussion

With this study, we have achieved radical C–$CF_3$ bond formation by the recombination of two radical species—one generated biocatalytically and the other in a chemical reaction. This method represents the first biocatalyst-dependent trifluoromethylation of organic compounds, especially unprotected phenols, giving access to building blocks that were not accessible as major products by any other method described before. Moreover, the methods display a high functional group tolerance, allowing the conversion of aldehydes, esters and ketones without decomposition, which makes this method suitable for late-stage trifluoromethylations. The method proceeds under mild reaction conditions with high regioselectivity.

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

## Acknowledgements

E.B. received funding from the European Commission by a Marie Curie Actions-Intra-European Fellowship (IEF) in the project 'BIOCASCADE' (FP7-PEOPLE-2011-IEF). N.R. has been supported by the Austrian BMWFJ, BMVIT, SFG, Standortagentur Tirol and ZIT through the Austrian FFG-COMET-Funding Program. Support by NAWI Graz and COST Action CM1303 Systems Biocatalysis is acknowledged. V.R. thanks the Austrian Science Fund (FWF) for an 'Erwin-Schroedinger' Fellowship (J3292).

## Author contributions

R.C.S. and W.K. conceived, designed and supervised the project. E.B. performed chemical control reactions, V.R. photometric assays, R.C.S. and N.R. analytical optimization studies. Preparative scale experiments and interpretation: R.C.S. and V.R. W.K. performed computational experiments. R.C.S., K.N.H. and W.K. wrote and edited the manuscript. All authors discussed the results and commented on the manuscript.

## Additional information

**Competing financial interests:** The authors declare no competing financial interests.

