## [Peer Review File · Nature Communications]

Reviewer #1 (Remarks to the Author):

Simon et al. use a laccase enzyme to generate a phenol radical cation through one-electron transfer and then trap this intermediate with $\cdot\text{CF}_3$ (generated non-enzymatically) to afford direct addition of CF_3 to unfunctionalised positions of unprotected phenols. The method is complementary to existing non-enzymatic methods for aryl-trifluoromethylation and unlike the existing CH activation approaches there is no need for a strong donating or coordinating group. Given that the process is dependent on the laccase, it is the first demonstration of a biocatalyst-dependent trifluoromethylation, which could be of considerable interest, particularly if the substrate scope could be broadened.

Both elements of these chemobio-transformations (i.e. laccases, Langolis' and Baran's reagents) are well known, although there are few examples where laccases are used for synthetically useful reactions. The novelty stems from bringing the two together and using the biocatalyst to obtain regioselectivity that may not be possible using the existing methods.

A small number of substrates are shown to undergo trifluoromethylation with moderate yields. While in some cases mixtures of regioisomers are formed (table 2 entries 4 and 5), it is clear that there is a preference for one regioisomer (4:1 to 10:1). Entry 6 does give a single regioisomer (2f), however, and the authors do a good job of explaining the regioselectivities by calculating transition state energies. The mechanism they propose based on these calculations is also plausible.

In order to put their work in context, they compare their chemobio-transformation with existing non-enzymatic trifluoromethylation strategies and they show that their method is complementary providing different regioselectivity.

Overall I am supportive of publication. Trifluoromethylation is a very important transformation, particularly in pharma. While there have been a number of synthetic methods developed to affect such transformations, this is still challenging chemistry. The possibility of using an enzyme to trifluoromethylate phenols is novel and will be of considerable interest to those in the synthesis and biocatalysis communities in academia as well as industry.

There are a few issues that the authors should address to improve the paper:

- 1) The six examples provided in table 2 do provide a proof-of-principle. However the substrates are structurally similar and the paper would benefit from some more examples including greater structural diversity - e.g. phenols with different substituents, biaryls, naphols, anilines... (within the timescale for publication).
- 2) Is the comparison with literature methods (Scheme 3) fully exhaustive? There are other methods such as copper/Togni reagent etc. It would be more convincing if the chemobio-transformations were compared with a wider range of non-enzymatic approaches (within the timescale for publication).
- 3) The yields are moderate at the moment. What is the reason for this and why is the yield of entry 6 (2f) based upon recovered starting material?
- 4) Compound characterisation is thorough. However there are no HRMS reported for any compound, which would normally be required.

Reviewer #2 (Remarks to the Author):

In my view, this submission would be appropriate for consideration for dissemination in Nature Communications, subject to the modifications discussed in the detailed review (attached).

Reviewer #3 (Remarks to the Author):

This communication describes a new protocol for the introduction of the CF₃ group into unprotected phenols, based in the recombination of two radical species (phenol radical cation and CF₃-radical). The phenol radical is formed by a biocatalyst (laccase using O₂ as oxidant), whereas the CF₃-radical is generated from trifluoromethanesulfinate (TFMS) salts and tBuOOH. DFT calculations have been performed to account for the regioselectivity of the radical recombination process.

Although Na-TFMS has already been used as CF₃• source for C-H trifluoromethylation of heterocycles (ref. 38), the present approach is innovative regarding both the biocatalytic generation of the other radical and the substrates covered (unprotected phenols). The mechanistic study dealing with the radical recombination is sound. The paper is potentially suitable for Nature Communications, but an important aspect of this study, the generation

of CF₃• from TFMS and tBuOOH requires further elaboration. The authors state initially that the two radical are formed via two independent pathways, but later on they state that Cu(I) present in laccase is required to react with tBuOOH to give tBuO•, which in turn reacts with TFMS to generate the CF₃-radical. The authors should deeply explore this half of the trifluoromethylation, both experimentally and theoretically, to account for the formation of the CF₃-radical.

Reviewer #4 (Remarks to the Author):

This is an excellent paper which describes an original cascade combining a laccase with a radical trifluoromethylation.

Considering the broad interest in trifluoromethylation and the lack of broad synthetic methods, this is a very significant contribution which will attract broad interest.

The paper is carefully designed and the control experiments and calculations confirm that the hypothesis that leads to trifluoromethylation of phenols is indeed valid

The regioselectivity issue is an interesting one.

While symmetric tetrasubstituted arenes yield a single regioisomer, less substituted phenols tend to afford meta-substitution products. In all cases however, the para position of the phenol is blocked by a carbonyl-bearing moiety. What happens if this group is absent? Does the phenol dimerize? What other functional groups are tolerated by this cascade?

The authors state the two independent pathways operate in a cooperative fashion: what do they mean with this?

Overall, an outstanding paper which requires only very minor revisions before acceptance

Reviewers' comments:

Reviewer #1 (Remarks to the Author):

Simon et al. use a laccase enzyme to generate a phenol radical cation through one-electron transfer and then trap this intermediate with $\cdot\text{CF}_3$ (generated non-enzymatically) to afford direct addition of CF_3 to unfunctionalised positions of unprotected phenols. The method is complementary to existing non-enzymatic methods for aryl-trifluoromethylation and unlike the existing CH activation approaches there is no need for a strong donating or coordinating group. Given that the process is dependent on the laccase, it is the first demonstration of a biocatalyst-dependent trifluoromethylation, which could be of considerable interest, particularly if the substrate scope could be broadened.

Both elements of these chemobio-transformations (i.e. laccases, Langolis' and Baran's reagents) are well known, although there are few examples where laccases are used for synthetically useful reactions. The novelty stems from bringing the two together and using the biocatalyst to obtain regioselectivity that may not be possible using the existing methods.

A small number of substrates are shown to undergo trifluoromethylation with moderate yields. While in some cases mixtures of regioisomers are formed (table 2 entries 4 and 5), it is clear that there is a preference for one regioisomer (4:1 to 10:1). Entry 6 does give a single regioisomer (2f), however, and the authors do a good job of explaining the regioselectivities by calculating transition state energies. The mechanism they propose based on these calculations is also plausible.

In order to put their work in context, they compare their chemobio-transformation with existing non-enzymatic trifluoromethylation strategies and they show that their method is complementary providing different regioselectivity.

Overall I am supportive of publication. Trifluoromethylation is a very important transformation, particularly in pharma. While there have been a number of synthetic methods developed to affect such transformations, this is still challenging chemistry. The possibility of using an enzyme to trifluoromethylate phenols is novel and will be of considerable interest to those in the synthesis and biocatalysis communities in academia as well as industry.

There are a few issues that the authors should address to improve the paper:

1) The six examples provided in table 2 do provide a proof-of-principle. However the substrates are structurally similar and the paper would benefit from some more examples including greater structural diversity - e.g. phenols with different substituents, biaryls, naphols, anilines... (within the timescale for publication).

Within the three months until the deadline for submission of the revision we tested various compounds and selected two addition compounds for scale up to identify the product. It turned out that also another functional group like nitrile moiety is tolerated. Additionally a substrate without a para substituent was scaled up. This data was added to the manuscript as well as information about substrates which were not converted at all under the reaction conditions investigated or which resulted in complex product mixtures.

Moreover, the reaction system tolerated ketone-, ester-, aldehyde as well as nitrile-functionalities emphasizing the mildness of the reaction. Interestingly nitrogen containing substrates like indol or 4-aminoacetophenone were not converted at all under the reaction conditions investigated, while other substrates like sesamol, 5,6,7,8-tetrahydro-2-naphthol, 2-naphthol or meta-dimethylamino acetophenone resulted in complex product mixtures.

Table 2. Scope and functional group tolerance of the biocatalytic trifluoromethylation of unprotected phenols employing Baran's zinc sulfinate or Langlois' reagent in combination with a laccase.

entry	product	Zn(SO ₂ CF ₃) ₂		NaSO ₂ CF ₃
		Isol. [%]	yield [%]	Isol. yield [%]

...

7		56.7	n. p.
8		33.1	n. p.

a ~4% double trifluoromethylation at C2 and C6 was observed; b isolated yield of the pure regio-isomer **2d-meta**; c overall isolated yield of both regio-isomers **2e-ortho** and **2e-meta**. n. p. = not performed.

2) Is the comparison with literature methods (Scheme 3) fully exhaustive? There are other methods such as copper/Togni reagent etc. It would be more convincing if the chemobio-transformations were compared with a wider range of non-enzymatic approaches (within the timescale for publication).

As an extension for the literature method the Togni reagent was also tested for substrate **1a**. The text and Figure 6 were consequently extended.

As a third method the Togni reagent^{32,46} was employed for substrate **1a** (Fig. 6, bottom); although the substrate was completely converted, product **2a** was only a minor product (8%) while two non-identified main products were detected, which did not contain any CF₃-group.

Thus, the laccase trifluoromethylation reported here is clearly complementary to literature methods tested.

Figure 6. Trifluoromethylation methods for electron rich arenes from literature tested for comparison with the here presented laccase/*t*BuOOH concept. One method (top) involved silver as metal while the second and third methods (middle and bottom) were metal free.

3) The yields are moderate at the moment. What is the reason for this and why is the yield of entry 6 (2f) based upon recovered starting material?

To be concise the yield upon recovered starting material was removed and the table adapted. The isolated yields we report range for the Baran reagent between 31% and 62%, which is in the range of various other (non enzymatic) publications; E.g.in Baran's nature paper the yields range from 35 to 79%, having 89% in one single case and in the PNAS paper the yields range in general between 33 to 78% (a single best value is 96%). Higher yields may be obtained by controlling an undesired side reaction, namely the aryl-aryl coupling of two phenol radicals avoiding di- or oligo/polymerization; this may be achieved ensuring that the CF₃ radical is present in excess (e.g. by additional electrochemical reactions, see ref. 42).

4) Compound characterisation is thorough. However there are no HRMS reported for any compound, which would normally be required.

We have now performed HRMS measurements. The additional data was added to the SI. At "General information" details about the equipment and settings were added:

HRMS were recorded on an HPLC-TOF-MS system consisting of an HPLC [Agilent 1260 Infinity Series; Injection: 0.1 µL; eluent: isocratic, 20% H₂O, 80% (90% ACN with 10%H₂O(0.1% 5 M Ammoniumformate)), flow: 0.3 mL/min] and Agilent 6230 TOF LC/MS with APCI measuring in the negative mode and the following settings: gas temp. (N₂): 350 °C; vaporizer: 375 °C; drying gas: 10 L/min; nebulizer: 40 psig; fragmentor: 50 V; skimmer: 65 V; OCT 1 RF Vpp: 750 V; Vcap: 3500 V; corona: 22 µA; nozzle voltage: 1100V; reference masses: 966.0007250; acquisition: 150-1,100 m/z; 1 spectra/s.

For the different products the results were added:

For **2a**: HR-MS: $m/z = 263.0537$ [(M-H)⁻] (calcd.: 263.0537).

For **2b**: HR-MS: $m/z = 249.0378$ [(M-H)⁻] (calcd.: 249.0380).

For **2c**: HR-MS: $m/z = 293.0643$ [(M-H)⁻] (calcd.: 293.0642).

For **2d**: HR-MS: $m/z = 233.0428$ [(M-H)⁻] (calcd.: 233.0431).

For **2e-meta**: HR-MS: $m/z = 263.0539$ [(M-H)⁻] (calcd.: 263.0537).

For **2e-ortho**: HR-MS: $m/z = 263.0537$ [(M-H)⁻] (calcd.: 263.0537).

For **2f**: HR-MS: $m/z = 203.0327$ [(M-H)⁻] (calcd.: 203.0325).

Reviewer #2 (Remarks to the Author):

In my view, this submission would be appropriate for consideration for dissemination in Nature Communications, subject to the modifications discussed in the detailed review (attached NOW inserted below).

In this submission, Kroutil and coworkers demonstrate that under appropriate conditions, the enzyme laccase can be used to regioselectively generate a C---CF₃ bond on appropriately decorated phenolic substrates. The best CF₃ donors appears to be the trifluoromethylsulfinate salts due to Baran and Langlois; particularly the former. This is a very creative and potentially useful new type of enzymatic transformation. The installation of trifluoromethyl groups on to aromatic centers is of great interest to the medicinal chemistry, synthetic chemistry and chemical biology communities. That this can be conducted under mild conditions, in chemoenzymatic fashion, should be of broad interest, and argues for publication of this results in a venue such as Nature Communications.

While the regiochemistry is high and the authors provide computational support for the observed regiopreference, the substrate scope appears quite limited, and it is not clear if the authors will be able to

expand the substrate scope to include fused aromatics, heterocycles, etc., perhaps with the assistance of saturation mutagenesis/"directed evolution" efforts. The authors could comment on this.

Following also the comment of the first reviewer additional substrates were tested and the scope of functional group tolerated was extended (e.g. nitrile group). We also added non-substrates or substrates leading to complex product mixtures. The substrate scope will depend always whether the laccase can oxidise the phenolic substrate; indeed protein engineering may help to adapt the redox potential and allow to transform other substrates.

Also, to enhance the paper, it is recommended that the authors add reference to the following examples of the use of laccase for C---C and C---O bond---forming reactions:

(1) "Laccase and Xylanase Incubation Enhanced the Sulfomethylation Reactivity of Alkali Lignin," H. Zhou; X. Qiu; D. Yang; S. Xie *ACS Sustainable Chem. Eng.* **2016**, 4, 1248–1254. DOI: 10.1021/acssuschemeng.5b01291

(2) "Chemoselective C4 Aerobic Oxidation of Catechin Derivatives Catalyzed by the *Trametes villosa* Laccase/1---Hydroxybenzotriazole System: Synthetic and Mechanistic Aspects" R. Bernini; F. Crisante; P. Gentili; F. Morana; M. Pierini; M. Piras" *J. Org. Chem.* **2011**, 76, 820–832; DOI: 10.1021/jo101886s

While paper (1) describes the increase of sulfomethylation reactivity by 33%, thus it reports a C-C bond formation, the second paper deals with hydroxylation via a mediator, which is out of scope of this paper. Thus only the reference for paper (1) was added as reference 38.

38. Zhou, H., Qiu, X., Yang, D. & Xie, S. Laccase and xylanase incubation enhanced the sulfomethylation reactivity of alkali lignin. *ACS Sustainable Chem. Eng.* 4, 1248-1254 (2016).

Greater discussion on mechanism, both additional potential justification(s) for the mechanism proposed and consideration of alternative pathways is probably warranted. The authors propose a radical recombination mechanism for this C---C bond---forming reaction. From their discussion, one presumes they are positing that this an active site--directed process.

There seems to be a misunderstanding: we do not assume that the regioselectivity is active-site directed, because this would be in sharp contrast to the calculated ratios of the energies of the transition states leading to different regio-isomers. To clarify this we added an additional sentence:

Since the energies of the transition states reflect the observed regioselectivity, the bio-trifluoromethylation is mainly not active site directed.

However, from what I could see, the authors present no real evidence of the generation of CF₃---radicals from Baran's salt under the conditions of their enzymatic incubations. As a part of this discussion, it is suggested that the authors explicitly discuss CF₃---radical formation and instances in which EPR signatures of this species have been observed, for example:

"Rotation Dynamics Do Not Determine the Unexpected Isotropy of Methyl Radical EPR Spectra" N. P. Benetis; Y. Dmitriev; F. Mocci; A. Laaksonen" *J. Phys. Chem. A* **2015**, 119, 9385---9404 DOI: 10.1021/acs.jpca.5b05648

(4) "Radiation Induced Redox Reactions and Fragmentation of Constituent Ions in Ionic Liquids. 1. Anions" I. A. Shkrob; T. W. Marin; S. D. Chemerisov; J. F. Wishart " *J. Phys. Chem. B* **2011**, 115, 3872---3888 DOI: 10.1021/jp2003062

It is true that we did not provide evidence for the generation of CF₃-radicals; however, in previous publications (e.g. ref. 39) this has already been proven by EPR studies in aqueous solution for CF₃SO₂Na and tBuOOH.

To clarify this a sentence was added:

Thus, the Cu(I) of the laccase can be oxidized by *t*BuOOH, which leads to *t*BuO• as previously reported;³¹ the latter reacts with the CF₃SO₂⁻ to set free the CF₃-radical as proven elsewhere.³⁹

An alternative mechanism might involve direct formal trifluoromethyl anion transfer from Baran's salt to the activated substrate radical cation in the enzyme active site (see scheme below). Here the role of the oxidant would be to reoxidize the Cu(I) to Cu(II) at each stage. Absent evidence for the generation of long-lived CF₃-radical species under the conditions of this enzymatic transformation, it would seem prudent not to rule out such possibilities.

Alternative mechanistic suggestion

Alternatively, one might even consider capture of the CF₃ group via an enzymatic Cu--CF₃ species. This is somewhat reminiscent of Hartwig's trifluoromethylating reagent, though, of course, the Hartwig chemistry itself is quite different:

(5) "A Broadly Applicable Copper Reagent for Trifluoromethylations and Perfluoroalkylations of Aryl Iodides and Bromides" H. Morimoto; T. Tsubogo; N. D. Litvinas; J. F. Hartwig *Angew. Chem. Int. Ed.* **2011**, 50, 3793---3798, DOI: 10.1021/jp2003062

We highly appreciate the efforts of the reviewer to suggest alternative mechanisms and to join the discussion. Although our reported mechanism looks simple, we spent more than one year discussing different mechanisms and consuming a lot of paper for variations. The two mechanisms suggested by the reviewer can easily be ruled out, because if they would be true the trifluoromethylation would also work in the absence of tBuOOH, since laccases can regenerate Cu(II) also just at the expense of O₂ (which is always present under the conditions employed). This case was ruled out with the blank experiment in Table 1, entry 4. Our mechanism is satisfying all our observed data, blank reactions and calculations. To elaborate

more on the mechanism addressing the points raised by the reviewer a sentence for the mechanism was revised and extended:

As shown in the initial experiments, trifluoromethylation only occurred in the presence of laccase and TFMS/*t*BuOOH (Table 1, entry 1); the phenol starting material did not react with TFMS/*t*BuOOH (Entry 2) nor did the phenol radical cation (formed by laccase and O₂ present) react with TFMS (Entry 4).

Apparent Typo: Under "Functional group tolerance" the authors state that they have solved the X-ray structure for compounds **2a-c**, but Figure 1 suggests that this should be **2b-d**.

The text was corrected:

In these cases, products **2a-c** with the CF₃-moiety *meta* to the OH were isolated with exquisite regio-control (entry 1-3) and for **2b** and **2c** also verified by x-ray crystallography (Fig. 3).

Clarification: For Table 1, entry 4 with a 23% conversation reported, please clarify what product was isolated in 23% yield.

No formation of product **2a** was detected (the same as for entry 3). The products were not identified (most likely dimers as stated for entry 3).

Reviewer #3 (Remarks to the Author):

*This communication describes a new protocol for the introduction of the CF₃ group into unprotected phenols, based in the recombination of two radical species (phenol radical cation and CF₃-radical). The phenol radical is formed by a biocatalyst (laccase using O₂ as oxidant), whereas the CF₃-radical is generated from trifluoromethanesulfinate (TFMS) salts and *t*BuOOH. DFT calculations have been performed to account for the regioselectivity of the radical recombination process.*

*Although Na-TFMS has already been used as CF₃• source for C-H trifluoromethylation of heterocycles (ref. 38), the present approach is innovative regarding both the biocatalytic generation of the other radical and the substrates covered (unprotected phenols). The mechanistic study dealing with the radical recombination is sound. The paper is potentially suitable for Nature Communications, but an important aspect of this study, the generation of CF₃• from TFMS and *t*BuOOH requires further elaboration. The authors state initially that the two radical are formed via two independent pathways, but later on they state that Cu(I) present in laccase is required to react with *t*BuOOH to give *t*BuO•, which in turn reacts with TFMS to generate the CF₃-radical. The authors should deeply explore this half of the trifluoromethylation, both experimentally and theoretically, to account for the formation of the CF₃-radical.*

We confess that our text about the two independent pathways was partly guided by our initial concept; however, as it turned out, the CF₃ radical formation depends on the *t*BuO radical which is most likely formed via the Cu of the laccase. Consequently the abstract was adapted to:

The method relies on the recombination of two radical species namely the phenol radical cation, generated directly by the laccase, and the CF₃-radical.

In the "reaction concept" the following sentence was changed:

The cooperative approach for the trifluoromethylation of phenols via C–C bond formation presented in this paper is based on the recombination of two radicals, namely a CF₃-radical and a phenol derived radical, whereby the two radicals are formed via two independent pathways in a cooperative fashion (Scheme 2).

To

The approach for the trifluoromethylation of phenols via C–C bond formation presented in this paper is based on the recombination of two radicals, namely a CF₃-radical and a phenol derived radical, whereby the two radicals are formed via two different pathways in a cooperative fashion (Fig. 2).

Since our explanation for the deduction of the mechanism was probably incomplete we added further explanations:

Thus, the Cu(I) of the laccase can be oxidized by *t*BuOOH, which leads to *t*BuO• as previously reported;³¹ the latter reacts with the CF₃SO₂⁻ to set free the CF₃-radical as proven elsewhere.³⁹

Reviewer #4 (Remarks to the Author):

This is an excellent paper which describes an original cascade combining a laccase with a radical trifluoromethylation.

Considering the broad interest in trifluoromethylation and the lack of broad synthetic methods, this is a very significant contribution which will attract broad interest.

The paper is carefully designed and the control experiments and calculations confirm that the hypothesis that leads to trifluoromethylation of phenols is indeed valid

The regioselectivity issue is an interesting one.

While symmetric tetrasubstituted arenes yield a single regioisomer, less substituted phenols tend to afford meta-substitution products. In all cases however, the para position of the phenol is blocked by a carbonyl-bearing moiety. What happens if this group is absent? Does the phenol dimerize?

Following the comment of the reviewer substrate **1h** was tested, which does not have a substituent in para to the phenolic OH (see also comments for reviewer 1). GC-MS analysis showed as a main product the di-trifluoromethylated product having the introduced groups in para and meta to the phenolic OH. Mono-CF₃-substituted products were in lower amounts. Dimerization was not detected under the conditions of reaction and analysis.

What other functional groups are tolerated by this cascade?

Testing substrate **1g**, it turned out that also a nitrile moiety is tolerated. For changes performed see comments at reviewer 1.

The authors state the two independent pathways operate in a cooperative fashion: what do they mean with this?

Our original concept was the two independent pathways and we took the term “cooperative” from the concepts of cooperative catalysis. Since our final reaction scheme does not fully fit in this concept, we now took out the term “cooperative” from the paper.

Overall, an outstanding paper which requires only very minor revisions before acceptance

Reviewer #1 (Remarks to the Author):

I was supportive of the paper at the outset and I am satisfied that the authors have provided a robust response to the questions that were raised by the reviewers. The additional data/experiments included in the revised manuscript and supporting information improve the quality of the paper and I recommend publication.

Reviewer #2 (Remarks to the Author):

We appreciate the efforts of the authors to explore the substrate specificity more fully, and clarify some of the mechanistic discussion. The work appears appropriate for publication in Nature Communications now; this is certainly an interesting application of laccase toward a synthetic transformation of contemporary interest and importance.

Reviewer #3 (Remarks to the Author):

I appreciate the efforts made by the authors to address most of the questions raised by the reviewers. The paper has been improved upon in this revised version. Regarding my concern, I would have preferred to find some additional calculations for the reaction between TMFS and tBuO., to account for the formation of CF₃• radical. Despite this, I consider the paper suitable for publication.